# Public Perceptions Regarding Genomic Technologies Applied to Breeding Farm Animals: A Qualitative Study

**DOI:** 10.3390/biotech10040028

**Published:** 2021-12-03

**Authors:** Francis Z. Naab, David Coles, Ellen Goddard, Lynn J. Frewer

**Affiliations:** 1School of Natural and Environmental Sciences, Newcastle University, Newcastle upon Tyne NE1 7RU, UK; francis.naab@bristol.ac.uk (F.Z.N.); david.coles@hazyrays.com (D.C.); 2Enhance International, The Bacchus, Elsdon, Newcastle upon Tyne NE19 1AA, UK; 3Agricultural Marketing and Business, Faculty of Agricultural, Life and Environmental Sciences, 515 General Services Building, University of Alberta, Edmonton, AB T6G 2H1, Canada; egoddard@ualberta.ca

**Keywords:** breeding, ethics, farm animals, focus groups, genomics, public attitudes

## Abstract

The societal acceptability of different applications of genomic technologies to animal production systems will determine whether their innovation trajectories will reach the commercialisation stage. Importantly, technological implementation and commercialisation trajectories, regulation, and policy development need to take account of public priorities and attitudes. More effective co-production practices will ensure the application of genomic technologies to animals aligns with public priorities and are acceptable to society. Consumer rejection of, and limited demand for, animal products developed using novel genomic technologies will determine whether they are integration into the food system. However, little is known about whether genomic technologies that accelerate breeding but do not introduce cross-species genetic changes are more acceptable to consumers than those that do. Five focus groups, held in the north east of England, were used to explore the perceptions of, and attitudes towards, the use of genomic technologies in breeding farm animals for the human food supply chain. Overall, study participants were more positive towards genomic technologies applied to promote animal welfare (e.g., improved disease resistance), environmental sustainability, and human health. Animal “disenhancement” was viewed negatively and increased food production alone was not perceived as a potential benefit. In comparison to gene editing, research participants were most negative about genetic modification and the application of gene drives, independent of the benefits delivered.

## 1. Introduction

Much of the world population is still dependent on animals as a source of protein [1]. Increasing demand is a consequence of increased populations, incomes, and urbanisation in both low- and high-resource countries [2]. At the same time, there is increasing societal concern about ethical issues associated with animal welfare standards [3], such as, for example, intensive production systems [4] and the application of novel technologies to enhance animal production [5]. As a result, there has been a considerable focus within scientific and policy communities on the application of novel genomic technologies to improve animal production systems and disease resistance in livestock [6], including *inter alia* genetic modification [5], gene editing, such as CRISPR-Cas9 [7], and the prospective application of synthetic biology [8]. At the same time, there is a body of evidence to suggest that public acceptance of the application of genomic technologies to animal production systems is nuanced by the type of genomic technology being applied (e.g., genetic modification versus gene editing), the intended outcome of the modification (e.g., improved animal welfare or increased profitability), and the target organisms used in the modification (e.g., mammals, birds, or fish) [9]. Differences in public attitudes toward applying gene editing to agricultural crops have been observed when compared to genetic modification, and conventional breeding technologies are applied to meet the same objectives. Kato-Nitta et al. [10] report that participants in their survey tended to have more favourable attitudes toward gene editing than toward genetic modification when applied to crops. Attitudes toward the use of gene editing in plant breeding appear less firmly entrenched than for genetic modification [11,12]. It is notable that some applications of gene editing are more acceptable than others. Yunes et al. [13] report, in a quantitative study of gene editing applied to cattle, low public acceptance overall. In cases where support was given, it was highly dependent on the type and purpose of the application proposed. Similarly, Busch et al. [14] indicate that their participants evaluated the application of gene editing to promote disease resistance in humans most positively, followed by disease resistance in plants, and then in animals, but considered changes in product quality and quantity in cattle as the most negative outcome of gene editing.

The aim of the research presented here is to understand public perceptions of the use of different genomic technologies in breeding farm animals used in food production, including ethical concerns linked to different technological outcomes. An extensive body of literature regarding public perceptions and other socio-economic aspects of genetically modified animals or other genomic technologies such as cloning applied to food production and other areas of application is available (see, inter alia [15,16,17]), although less research has been conducted in relation to gene-edited animals. It has been established that the way people perceive new (food) technologies, for example, in relation to potential risks and benefits, determines whether they accept the development and implementation [18]. Risk perception refers to people’s subjective judgments about the likelihood of negative occurrences, such as negative impacts on animal health or the environment, and is important in health and risk communication because it determines which hazards people care about and how they deal with them [19]. This includes communication about buying products produced using genomic technologies [20]. Risk perception is important to policy makers as the public may reject technologies that they perceive to be risky or unethical, independent of technical risk assessments provided by experts. It is important to note that research into risk perception reflects an objective analysis of public or consumer attitudes (Nuffield Council of Bioethics (2021) Genome editing and farmed animal breeding: social and ethical issues. NCOB, London, UK).

Advances in biotechnology have given rise to novel approaches to breeding farmed animals for human consumption. This includes, for example, breeding disease-resistant, healthier, and more productive animals (e.g., the case of CRISPR-Cas9 in pigs [21,22,23]), animal production systems with reduced environmental impacts, for example, in relation to greenhouse gas emissions [24], and producing animals more amenable to being managed in existing animal husbandry systems [25]. It is important to understand how citizens perceive the application of different genomic technologies in animal production systems, as they are unlikely to be adopted if there is societal opposition to their application, which is frequently underpinned by moral concerns [26]. Public acceptance may be linked to both the acceptability of the specific biotechnological process applied in the process of modification [9] and the developers’ reason for applying it. For example, public concerns about the acceptability of animal products have focused on different issues, such as animal welfare [27,28] and environmental and human health concerns [29,30].

The evolving legislative framework, the potential impact of public perceptions of risk, benefit, and ethical concern on this framework [31], and the extent to which consumer perceptions have contributed to the European Union’s regulations regarding genetic modification within animal production systems clearly indicate [20] that consumer acceptability has to be taken into account.

However, differences in perceptions and attitudes need to be assessed in relation to different types of genomic application. Variations in legislative frameworks between diverse regions also exist. For example, within Europe, CRISPR-Cas9 technology is regulated in the same way as genetic modification (GM) [32], whereas in the US the resultant product is not considered GM. Hence, there exists a fundamental difference in approach where the US focus is on the ultimate “product”, while the EU focus is on the “process” [33].

### Perceptions of, and Attitudes towards, Genomic Technologies Applied to Agriculture

While there is an extensive body of literature on public perceptions of and attitudes towards the genetic modification of plants, and to some extent to animals and micro-organisms, other areas of genomic science applied to food production, for example, using animals, have not been so extensively researched. However, there is some evidence that public attitudes to gene drives used in agriculture are, as for genetic modification, nuanced by moral concerns and associated attitudes [34]. Similar findings have been reported for agricultural applications of gene editing [35]. Generally, the focus of this body of research has tended to be on understanding public attitudes to biotechnological methods applied to plants. There is little research conducted in relation to some comparator technologies, for example, accelerated animal breeding, although there is some evidence that the public associate the latter with genetic modification [36].

The focus groups methodology was applied in order to (1) explore the attitudes of UK citizens towards some genomic technologies; (2) discuss and consider ethical dilemmas that may occur as a result of the use of genomic technologies in animal production systems.

## 2. Materials and Methods

Following ethical approval for the research (Newcastle University Ethics Committee, approval number 7235/2018), focus group discussions were used to initiate discourses between participants, allowing the researcher to decipher and moderate the divergent opinions [37]. Five (5) focus group discussions were organised, with 6–12 individuals in each group discussion. In total, 38 respondents participated, and the discussions took place between November and December 2018. Four focus group discussions were conducted in the city of Newcastle, and the fifth in a village in rural Northumberland. Each focus group discussion lasted between 50 and 70 min and was moderated by a trained researcher and an assistant. Saturation was reached during the fifth focus group discussion, with no further information being obtained.

### 2.1. Recruiting Participants

Initially, posters and flyers advertised for potential recruits on public notice boards. Respondents who expressed interest in taking part in the discussions were sent further information regarding the study’s purpose and informed that if they were selected to take part, they would receive a GBP 10.00 shopping voucher. Interested respondents were sent a brief socio-demographic questionnaire to complete (age, occupation, gender, educational background, nationality, and dietary preferences). People below the age of 18 years were excluded, and participants with different socio-demographic characteristics were randomly allocated across the four urban and rural groups (Table 1). The results of the initial pilot group were included in the main analysis as no changes were made following the pilot.

Four of the discussions were held within Newcastle University and the fifth focus group discussion was held in Elsdon, a rural village 40 km north of Newcastle.

### 2.2. Structure and Approach to Focus Group Discussions

Participants were briefed on the discussions and asked to sign consent forms. They were subsequently randomly allocated numbers with which they were identified in the discussion in order to anonymise responses. They were from that point only referred to by their gender, random number, and the focus group in which they participated (e.g., a female that received the random number 3, in focus group 1, would be identified only as (F3, FG1). All focus groups followed the same protocols, developed by the authors of this paper. Each focus group discussion was preceded by a PowerPoint presentation where the moderator presented an overview of the technological issues discussed in the focus groups in relation to biotechnology.

#### 2.2.1. Part I—Attitude to Different Genomic Technologies Applied to Animal Production Systems

Participants were provided with descriptions of various genomic technologies applied to animals used in food production systems. As a “warm -up” exercise, participants were asked to rate how ethically acceptable they viewed each type of genomic technology listed on a scale of 0–5, with 0 being unacceptable and 5 being entirely acceptable. The genomic technologies considered here were genetic modification (GM), structural genomics, functional genomics, conservation genomics, proteomics, and gene drive (see Table 2). Participants were asked to describe why they assigned the score given to that particular genomic technology, which then led to a group discussion of the various technologies about why the various technologies were or were not ethically acceptable. The aggregated results were provided as feedback to frame the discussion, but will not be considered further here because small sample sizes mean that statistical analysis is not appropriate.

#### 2.2.2. Part II—Relative Importance of Genomic Technologies Applied to Animal Production Systems

This section was designed to understand what participants thought about the different genomic technologies under consideration in relation to their potential impacts within society. Information was provided to participants linking some claimed potential benefits of the use of genomic technologies to breeding farm animals, including pigs and cattle.

#### 2.2.3. Part III—Ethical Dilemmas in the Use of Genomic Technologies

Given that ethical concerns are raised as an important societal barrier to the adoption of GM in animal production systems, this was further explored in relation to each of the gene technologies under consideration. An important ethical question in animal breeding is the consideration of whether the “naturalness” (or “*telos*” (The telos of an animal is defined as “its nature or ’beingness’”. Harfeld, J.L., 2013. Telos and the ethics of animal farming. Journal of agricultural and environmental ethics, 26(3), pp.691–709. In the focus groups, this was considered by participants as being linked to “naturalness” and so the two concepts are addressed together in the subsequent analysis and discussion.)) of animals should be preserved. The concept of naturalness was introduced to the groups, which led to their discussing the extent to which the animals themselves might ethically be adapted to their environment in order to promote and improve their welfare and facilitate their management. Both natural and unnatural methods of adaptation were discussed, including cases where some methods may lead to the “disenhancement” (Making an animal less sensitive to and more able to cope with adverse characteristics that may exist in its environment that may prove difficult to that animal species in its natural state, e.g., see Murphy, K.N. and Kabasenche, W.P., 2018. Animal disenhancement in moral context. NanoEthics, 12(3), pp. 225–236) of farm animals, and thus potentially affect their “*telos*”. This may reflect a more biocentric perspective, which is, in a political, ecological, and literal sense, an ethical point of view that extends inherent value to all living things (see, inter alia [38,39]).

In order to facilitate and catalyse the discussions, two scenarios were presented to participants. These are provided in Appendix A.

Each scenario was discussed with participants in relation to the acceptability of each of the solutions with regard to animal welfare and the *telos* of the animals and whether applying the different genomic technologies to achieve the same goals would be preferable. Finally, the moderator encouraged the discussions to consider more general attitudes and perceptions of participants towards using genomic technologies in breeding farm animals, focusing on the various ethical considerations.

### 2.3. Data Recording, Coding, and Analysis

All discussions were recorded and transcribed. Preliminary codes were developed from notes taken during discussions and debriefing sessions. Thematic analysis was subsequently conducted on the transcripts using NVivo, version 20, QSR International, Melbourne, Brevard County, Australia. Thematic analysis is a qualitative data analysis method that involves reading through a dataset (such as transcripts from focus groups) and identifying patterns in meaning across the data [40]. Coding was initially validated using the first focus group transcript and further discussed by researchers. During this process, further codes (and subcodes) emerged. Codes and sub-codes were then organised into themes and sub-themes (Table 3). The remaining transcripts were then coded, and the coding scheme amended if additional themes emerged during the analysis. To maintain the anonymity of participants, the audio records were destroyed once transcription was completed and validated, in line with ethical requirements for participant anonymisation required by General Data Protection Regulation (GDPR) requirements. This may have limited subsequent reanalysis of participants’ affective responses, but was appropriate given data protection regulation applied at the time of data collection.

The researchers discussed the various themes that emerged from the analysis. The quotes used in the text represent the emerging themes and the divergent opinions held by participants.

## 3. Results

All focus group discussions were highly interactive, with all participants engaging in the discussion. This reflects the level of interest, understanding, and knowledge of the participants relating to the subject area. In general, participants were open-minded and frequently challenged the views of other participants. Most of the participants agreed that there was a need to conduct genomic research into animal production. However, participants expressed divergent views concerning how the information should be applied. Some participants expressed specific cultural and ethical views about the use of various genomic technologies in animal breeding. In contrast, others were more positive about using genomic technologies if those technologies were appropriately regulated, and if societal preferences and priorities for technological innovation were taken into account. Some participants argued that genomic technologies could improve animal welfare and/or improve environmental sustainability. The overall conclusions of the discussions were broadly consistent across the five groups and reflected in the various themes presented in this research.

### 3.1. Attitudes towards to the Use of Genomic Technology in Animal Production Systems

The first part of the discussions involved participants assigning scores to six genomic technologies and then discussing their results with the rest of the group. The means of the scores were then calculated and used to complement the discussions that followed. Participants tended to agree about the acceptability of proteomics, conservation genomics, structural genomics, and functional genomics (see Table 2).

Participants were less positive about applying genetic modification and gene drives in breeding new traits in farm animals for human food production. Proteomics was rarely mentioned in the discussions, and participants were more confident in discussing conservation genomics and structural genomics. Participants associated structural genomics with traditional selective breeding.
*…It’s like selective breeding except you have more revision and knowledge to see what are actually selectively breeding towards…if you are looking at the actual genes you know what you are aiming for, you don’t have those mistaken ones, you have the good ones…*(M, FG2)

However, a few participants, after clarifying the meaning of both structural and functional genomics, were negative towards these applications and/or the use of information from them.
*We’ve always done it for years, but we’ve just modernised it to an advanced state that now threatens our very existence.*(M, FG1)

Some participants indicated that natural selection is a natural process, but that this was not the case for all applications of genomic technologies. This line of reasoning represented a common thread through subsequent discussions. Participants held that understanding genomic structures and using that information for selective breeding of farm animals would benefit farmers. However, a majority of the participants expressed the view that such breeding processes should be “natural”, for instance, by identifying, through genomic analysis, and then selecting pigs with a desirable trait to produce pigs with traits “useful” for supply chain requirements. Most participants expressed more negative opinions about genomic technologies perceived as “artificial” (especially GM and gene drives), especially for the development of animals with characteristics that could have unintended negative effects, with potentially severe consequences for human health and wellbeing as well as for the animal species concerned.
*I just think it could be dangerous, like getting rid of a gene or modifying it. Like what they did to mosquitoes, that’s good, but if you start doing to the animals we eat, it’s hard to model interactions as in the rest of the ecosystem, so it might result in something negative.*(F, FG2)

### 3.2. Imposing a Global Control System

Some participants expressed the view that there are potential benefits from the application of *all* the genomic technologies discussed if the route to application was cautious, precautionary, and appropriately regulated. However, they also recognised the difficulty of establishing an appropriate and consistent regulatory framework.
*At first I indicated we should use these technologies with moderation, but how can we moderate their use, on a global scale, it has become increasingly difficult.*(F, FG 3)
*So where do you draw the line when you start doing that?*(F1, FG4)

Some participants did not believe that individual and country-level controls were adequate to moderate potentially extreme or maleficent use of genomic technologies:
*……there should be a global thing, ……there should be a global kind of system of policing, and you know the sort of ethical side.*(F1 and M2, FG5)

Participants expressed the need for industry and national regulation to introduce a system of governance that includes regulations based on ethical considerations as well as risk issues.


*……. that code of conduct, that code of ethics, so there should be some mechanism by which they [industry and national governments] have to be held to accountable to.*
(F1, FG5)

This was linked to the need to apply universal governance:


*I think the most important part is to give the universal limits, I think that’s the reason why the government exists, to set limits for some of these things.*
(F3, FG4)

### 3.3. Applications for Health Versus Food

Participants agreed that there was a need to conduct research in these areas using animal genomic technologies. Many participants indicated that they thought there was the need to use genomics for fundamental research, particularly functional genomics, structural genomics, conservation genomics, and proteomics, in order to understand the nature of organisms.
*I feel like there’s a line between the research and the product… I find important to do research because I feel that is the only way we would be able to understand anything after.*(M, FG 1)

The development and application of genomic technologies and their potential benefits were also regarded as important by some participants. While participants held generally negative views in relation to applying the genomic techniques to animals for food production purposes, participants were more positive about the application of genomic technologies for medical and veterinary research and to health care.
*If we’re all scared of manipulation, I’m not sure we would have gotten treatment for some of the diseases we have”*(F3, FG3)
*…we already know about the CRISPR/Cas9 technology and it’s currently used in cancer technology and the treatment of cancer*(F4, FG4)
*…I think its uses in disease control, for example if this became an alternative to badger culling……*(M, FG2)

Some study participants expressed reservations about the use of genomics and genomic technologies in this context, particularly if they involved the insertion of human genes in animals for the purposes of xenotransplantation. This view was sometimes expressed through the lens of religion.
*I think that these [genomic] technologies in general] would be more suited for sustaining food production… but I don’t agree too much with applying to human health, I feel like the human health, I think it’s worse, and for me my religion doesn’t permit that”*(M, FG 1)

#### 3.3.1. Fear of the Unknown, Novel or Unanticipated Outcomes in Animal Production Systems

The fear of the “unknown”, which underpinned some participant concerns about using genomic technologies applied to animals for food production, was mainly linked to discussions on genetic modification and gene drives. This may be related to concerns about cross-species genomic technologies or perceptions of uncontrollability.
*Mixing species is just mind blowing, it frightens me just the thought of mixing genes from different species*(F1, FG5)
*…it can propagate through the population-what if you were wrong, then you screwed it up basically*(F1, FG3)

Many participants expressed a preference for the application of genomic technologies to conserve animal species compared to other potential benefits that could be generated from the use of such technologies. This was not particularly related to farm animals but related more generally to both domestic animals and those living in the wild. This is also consistent with views from discussions which suggest that participants were more open to the study and analysis of the genomes of animals and plants, which, in most cases, can be achieved through conservation genomics, proteomics, structural genomics, and functional genomics, but were highly sceptical about genetic modification and gene drives. However, many participants displayed some concerns about how knowledge gathered from the study of animal genomes might be applied in the future.

#### 3.3.2. Perceptions of Benefits Associated with the Use of Genomic Technologies Applied to Animal Production Systems

For most participants, the beneficial aspects of genomic technologies linked to outcomes that would improve the lives of animals were considered to be the most important, for example, in relation to conservation genetics. These included:

#### 3.3.3. Animal Health and Welfare

Discussion about animal health and welfare was an important topic for most participants and was closely linked to the concept that animal welfare equates to animal health and disease control.
*…animal health …control of animal welfare.*(F1, FG3)
*…improved animal health which I think it probably crosses over to animal welfare, that idea of any common diseases, approach as many as you can sort of help identify you breed out diseases*(M3, FG5)

While some participants perceived animal health as being distinct from animal welfare, the majority viewed animal health as part of animal welfare. Thus, genomic technologies were viewed to be helpful and important if they could be used to improve animal health and welfare. The majority of participants viewed animal welfare as an important consideration when using genomic technologies. Following the presentation of the case studies, some participants disliked the idea of using genomic technology to cause changes in animals. One participant described it as an “obvious violation” (F, FG1) of the rights of these animals and, to another, “absolutely unacceptable,” (M2, FG2) or “inhumane”.

However, a few participants disagreed, reasoning that:
*… all animals were created in the perfect form, heat, drought, water resistant etc.… you can find each animal is created in its form and place”.*(M5, FG2)

For some participants, genomic technologies were seen as advantageous to animal welfare and hence production.
*I don’t think breeding to make them gentler and less aggressive is slightly less unethical [laughter]*(M, FG4)

Some participants indicated that they felt that animal welfare was only used as a justification for genomic research insofar as this was a scientific bridge to improving human health.
*…lot of the things we are talking about we’re saying it’s better for animals, they are not necessarily better for animals, they are better for us to get something out of the animals and I’m not sure if I am totally comfortable with that*(F1, FG3)

Other participants indicated that, independent of the reasons why farmers might use genomic technologies (for example, to lower costs of production, increase productivity, and to improve environmental sustainably to mitigate the impacts of not using optimum animal husbandry practices), the welfare of animals should always be considered and given a high priority. Participants who held this view suggested that seeking the best welfare conditions for animals ultimately will lead to all other benefits that farmers may seek.
*When the animal is feeling better, when the animal is feeling natural, when their wellbeing is enhanced then they will be more productive in the end, if they lived in a natural environment the food would be safer, they would feel more motivated and healthy*(F1, FG4)
*But basically you have got to link your low cost with your animal welfare, the two have got to come together*(M5, FG5)

Some participants expressed the view that animal welfare problems were a consequence of human actions and that the conditions associated with animal husbandry should be changed to accommodate animal welfare needs, rather than developing new technological approaches to addressing animal welfare within these systems.
*“If people want to continue eating meat, then surely they should change the environment that these animals are supposed to live in and not genetically modifying the animals to endure the conditions”*(F2, FG2)

#### 3.3.4. Safer Human Food and Health

Some participants emphasised the importance of the application of genomic technologies to improve human food security. In this context, some participants expressed concern about which genomic technologies should be applied to food production and the need to prioritise improved food safety over other beneficial impacts.
*… Safer human food should be first [most important] because if you want for example a lot of the health problems we encounter nowadays can be traced to the food we eat…having a safe diet can help cut the risk of certain diseases like cancer*(F3, FG4)

#### 3.3.5. Environmental Sustainability

Participants indicated that environmental and ecological problems and factors linked to climate change could justify the application of many genomic technologies if their application mitigated these. For some participants, environmental sustainability was viewed as the most important potential impact of genomic technologies, leading to greater productivity, safer human food, improved animal health and welfare, and the preservation of ecosystems. However, one participant considered that the motives driving investment in genomic technologies were driven only by the financial interests of corporations and individuals.
*The main thing and my fear is the money culture, you have separate ambitions and rules, and it can be an issue. I think it’s disgusting to try to genetically modify the horrible life of an animal just for money, and it’s nothing to do with global warming.* (F2, FG2)

#### 3.3.6. Low Cost and Greater Productivity

The use of genomic technologies to improve productivity was viewed as a “game changer”. Lower production costs and increased productivity associated with genomic technologies were viewed as important by some discussants, particularly in relation to global population growth.
*I think with technology, we can produce more at very low cost to feed the increasing human population.*(M1, FG5)

Not all participants, however, held this view. Some associated the use of genomic technology with financial incentivisation on the part of industry stakeholders. The view was expressed that there is enough food to feed the world population, but food insecurity is driven by inequitable food distribution of food and food waste in supply chains.
*…For me, I don’t think there is scarce food in the world, there is abundance of food, it is a problem of the distribution of the food that is causing all the food insecurity in the world”*(M, FG1)
*“The whole thing that screams at me …it says it’s all to do with lower cost and greater productivity…meanwhile the only reasons is to financially drive us to where and what we don’t need.*(M1, FG5)

#### 3.3.7. Naturalness and Temperament

Participants were generally not in favour of applying any genomic breeding techniques that resulted in the modification of the temperament or *telos* of animals. There was some difference of opinion amongst participants as to whether animal “naturalness” or animal temperament were more important in this regard.
*Naturalness, that’s highly important … the animals should be living their natural lives in a as close to it as possible…*(M, FG1)
*Temperament is slightly more important than naturalness, because by the time you have been domesticating animals for ten thousand years a lot of the naturalness [is lost].*(M2, FG2)

## 4. Ethical Concerns

The specific ethical scenarios considered during the discussions focused to a considerable extent on participant opinion on the use of genetic modification. The common themes that were consistent across the focus groups included animal rights and welfare, *telos*, access to more extensive conditions in which animals could be reared, and due consideration of alternatives to the use of genomic technologies.

### 4.1. Telos (Naturalness)

Many participants discussed the importance of maintaining the “naturalness” or *telos* of farm animals. The use of genomic technologies that completely change the natural characteristics, or attributes perceived to be natural, of farm animals was unacceptable. The view was expressed that the phenotypical features of animals were integral to the nature of the animal and existed for a reason.
*…While chickens use their eyes to see and beaks to feed, pigs wag their curly tails as a result of emotional expression.*(M, FG1)
*it’s disgusting to …. remove beaks, eyes or tails of animals or do anything that will make them look less animals*(F2, FG2)

Any changes in the phenotype of animals, unless it was for welfare and animal health which ultimately led to better productivity, were viewed to be unethical. Breeding to change the temperament of animals was seen by the majority as a violation of their fundamental nature and “natural rights”, and thus was not considered welfare-driven.
*In terms of naturalness, the aggression might be useful to the pig, it might be their nature to be aggressive.*(M, FG1)

Participants who expressed concerns about genomic technologies in animal welfare also indicated that these concerns also had an ethical basis. Alternative futures that did not involve genomic technologies were described.

#### Reduced Meat Consumption

Some participants emphasised the need to reduce meat consumption. Other participants expressed the view that they were in favour of reduced meat production while giving livestock appropriate space to enjoy their natural habitats. These discussants claimed that increased societal demand for meat is driven by increasing meat supply, which makes the meat very cheap. This, in turn, triggers the need for the introduction of genomic technological innovations.
*I think people are going to have to get used to the fact they have to pay more for their meat, it’s too cheap…**… In an ideal world we would all eat less meat, and we would have much higher welfare chickens*(F1, FG3)

### 4.2. Overall Concerns about the Use of Genomic Technologies in Animal Production Systems

#### 4.2.1. Use of Genomic Information and Technology

While there was a general agreement that the study of the structure and functions of the animal genomes was important, some participants had concerns about the ways in which the resulting information might be used, and that genetic information should only be used to promote animal health and welfare.
*…I have no problem with that, it’s just doing DNA analysis and obtaining information, it’s what you do with that information that is potentially disturbing.*(F, FG 3)
*Any technology that doesn’t seek to improve animal health, helping prevent diseases, or identify and cure ailments in animals and lead to higher animal welfare…in my opinion is not good to us…*(M2, FG4)

Other participants viewed genomic technologies as a way of improving productivity to meet the food security requirements of our ever-increasing global population, while to others, these technologies offered ways of protecting the environment and ensuring the existence of endangered species.

#### 4.2.2. Motivation by Financial Interests

Participants expressed concerns about the motives of industry actors that drive the use of genomic technologies. Some participants believed that farmers and producers are motivated by profit alone. Others were concerned about the role of patent rights that have been generated from the use of genomic information and technology.
*… It is just the huge businesses which will take over and then becomes another capitalist kind, you know where that transition occurs.*(F2, FG5)
*… my fears of the money culture [referring to financial interests of corporations] …*(F, FG2)

## 5. Discussion

The research results indicated that, although additional information and clarification had been provided throughout the focus groups, there remained a general lack of participant differentiation between the different genomic technologies applied to farm animals. Participants expressed a preference for “non-invasive” technologies where no genetic modification or editing was applied, but where technological innovation was directed instead towards mapping existing animal genomes and used to selectively breed for desirable traits. Ethical concerns were frequently expressed about the technological processes being applied, particularly in relation to the extent to which such processes were perceived to be different from “natural” breeding techniques, and also in relation to the objective of the application, such as animal health and welfare or environmental protection, which tended to be viewed as more ethical than applications that increased yield or economic value within supply chains. However, exceptions to this increased ethical acceptability were those genomic applications that increased animals’ tolerance of intensive production systems or lower welfare standards. This does not align with the argumentation proposed by Thompson [41], where it is proposed that genomic disenhancement, which changes the *telos* of animals so as to enhance animal welfare in intensive animal production systems, is acceptable. The results appear to reflect Thompson’s perspective that there are moral intuitions that militate against animal enhancement in the absence of strong philosophical arguments against it. Some ethicists argue that disenhancement may be a temporary measure to relieve animal suffering in environments that humans have created for them [42], although it has, in turn, been argued that it is better to address the conditions that give rise to poor welfare [43] (an issue also raised in the focus group discussions), or indeed rethink the concept of telos to address what “is important” to an animal [44]. Further research to understand differences between the ethical reasoning of experts and public representations of and beliefs about the same issues is required.

When provided with further information about the different technological innovations applied to animals, participants expressed the view that gene editing (where the cell’s genome can be cut at a desired location, allowing existing genes to be removed and/or new ones added, representing the precise and targeted alteration of a DNA sequence in a living cell) was preferred over and above genetic modification (involving the transfer of genetic material from another species) and the use of gene drives (which propagates a particular suite of genes throughout a population by altering the probability that a specific allele will be transmitted to offspring). The latter were associated with perceptions of unnaturalness and uncontrollability, as well as being perceived to increase risks to human and animal health, as well as the environment. The observation was that the focus group participants were more accepting of genomic technologies (for example, accelerated breeding) that did not result in “invasive” genetic changes but allowed a more rapid and precise strategy to genetic change based on the “observation” of genes rather than the introduction of artificial genetic changes. This suggests that it is not the concept of genetic change that is of concern to the public, but rather the technological mechanism by which it is achieved, and the extent to which this can be obtained using natural breeding techniques. As has been found in previous research studies, the perceived potential for unintended health and environmental impacts associated with genetic technologies (e.g., see [43,44]), and ineffective or contradictory regulatory mechanisms to control and mitigate these (e.g., see [45,46]), contributed to these concerns.

Participants recognised the transboundary nature of potential risks and ethical issues and suggested that, as well as the need to include and address ethical issues in the construction of regulations associated with genomic technologies applied to animals and their products, there was also a need to include these in transboundary regulatory systems, given that the risks and ethical issues also had transboundary implications. Ethical concern was expressed about the continued use of genomic technologies to further the development of existing intensive animal production systems in their current trajectory (for example, through applying these technologies to disenhance negative animal behavioural responses to such production systems), rather than mitigating the problems by reassessing production system structures and regulation, and so improving animal health and welfare through changing current practices. Notably, many participants perceived that the application of some or all genomic technologies applied to animal production systems was, in fact, unregulated at a global scale. This could be associated with the lack of trust that consumers have in research institutions, governance practices, and industry [5,45]. Financial gain was perceived by many participants to motivate the biotechnology industry to use all genomic technologies in ways that are potentially detrimental to animal health and welfare (see also [46,47,48,49,50]). It was suggested that developing and communicating how governance systems work at local, regional, and international levels might reassure the public that good governance practices are being applied. Increased co-production involving all sectors of society, in the development of regulations, policies, and how these are applied and monitored, may increase societal trust in governance practices. However, it is important to note that many different perceptions and opinions are likely to be associated with different individuals and groups within the public, and understanding these differences is an important issue in relation to the co-production of policies.

The technologies considered in this research were all identified as having at least the potential of being used for breeding farm animals [51]. Those technologies, which were all viewed as a means of studying and accumulating knowledge about the genomes of organisms, were, however, viewed more favourably than those that were associated with structural changes to an animal’s DNA. Participants did not really differentiate between different types of genomic technology unless prompted to do so, including in relation to their ethical concerns. There was general acceptance of traditional selective breeding techniques, and from this accelerated breeding technologies were also considered acceptable, assuming established breeding techniques were still used. However, those technologies where some modification of animal genetic structures was involved were considered less acceptable, although this was more pronounced for genetic modification than for gene editing such as the CRISPR-Cas9 technology. This suggests that a different labelling approach may be required for genetically modified, as distinct from gene-edited, animal products, as the latter may be more acceptable to concerned consumers than the former, although such an approach is not accommodated within some legal frameworks. For example, CRISPR-Cas9 technology is currently regulated under the GMO regulations in Europe [52]. The proponents of modern gene editing techniques such as CRISPR-Cas9 argue that such techniques can be used to give additional or more complex types of genetic changes to those that would occur naturally. This should therefore be taken into account in legislative frameworks, for example, in the EU definition [53]. The main question that needs to be addressed is whether products developed using gene editing should be regulated on the basis of the process or the final products’ characteristics, or whether a hybrid approach should be taken (https://www.europarl.europa.eu/RegData/etudes/ATAG/2020/641535/EPRS_ATA(2020)641535_EN.pdf, accessed 10 March 2021). It is important that the technologies used to produce foods will be labelled on products in order to promote transparency in food systems and the availability of information for those who would like it.

Finally, the convergence on views of discussants about the use of genomic technologies and information to facilitate human health care and veterinary research could be linked to the increasing role of genomics in health care [52,53]. While noting the relative importance of genomic technologies in health care for both humans and animals, participants also suggested that there were potential unintended consequences of undesirable traits being passed to offspring as a consequence of genetic alteration.

## 6. Conclusions

This research suggests that the public are more positive about the use of (various) genomic technologies to study and accumulate genetic information about animals, including farm animals, and which inform and accelerate traditional breeding practices, compared to techniques that modify the genome of animals. There was more consensus regarding applications that improved information for conservation, environmental sustainability, health, and animal welfare, with the exception of animal “disenhancement”. Maintaining the “*telos*” of animals was important to study participants. The integration of societal preferences into regulations and labelling strategies may increase public trust in science and regulatory institutions, but further research in different cultural contexts and at scale is needed to enable a “co-produced” future regulatory landscape to be developed.

## Figures and Tables

**Table 1 biotech-10-00028-t001:** Summary demographics of focus group discussion participants.

VARIABLE	NUMBER (%)
**Gender**	
Male (M)	22 (58)
Female (F)	16 (42)
**Age groups**	
18–30	18 (47.4)
31–43	9 (23.7)
44–56	4 (10.5)
>57	7 (18.4)
Mean (age)	37.6
**Nationality**	
United Kingdom	23 (60.5)
European	3 (8)
Asian	4 (10.5)
African	7 (18.4)
Caribbean	1 (2.6)
**Employment status**	
Unemployed	1 (2.6)
Paid employment	15 (39.5)
Student	18 (47.4)
Retired	4 (10.5)
**Self-stated dietary preferences**	
Asian	1 (2.6)
None	27 (71)
Halal	3 (8)
Lacto-ovo free	1 (2.6)
Vegan	2 (5.2)
Vegetarian	3 (8)
Non-Vegetarian Hindu	1 (2.6)

**Table 2 biotech-10-00028-t002:** Areas of genomic technology discussed.

Type of Genomic Technology	Brief Description of Technology	Examples of Application to Animal Production Systems for Food Use
Genetic Modification	Changing the genetic makeup of cells, including the transfer of genes within and across species boundaries, to correct defects or produce improved and/or novel organisms.	Insertion into pigs of spinach gene to change body composition for better food production. Insertion of a modified gene to create animals resistant to heat stress.
Structural Genomics	DNA sequencing, sequence assembly, sequence organisation, and management and determination of the structure of every protein encoded by the genome.	Identifying animals with “desirable” genes, e.g., greater productive yield, better disease resistance.
Functional Genomics	Reconstruction of genome sequences to discover the functions of the genes together.	Identifying how genes interact to produce desirable traits, e.g., animal behaviour, health, and increase in productivity.
Conservation genomics	Use of genomic sequencing to better evaluate genetic factors key to species conservation.	Establishment of the size and health of a gene pool or genetic diversity of a population including preserving at-risk genotypes.
Proteomics	The large-scale study of the structure of proteins and what their function is and how they interact in animals.	Understanding of the function and regulation of genes, and how these participate in complex networks producing proteins and other biological agents controlling the phenotypic characteristics of a trait.
Gene Drive	Natural or genetically engineering the characteristics of a particular trait so that it dominates other traits and can propagate throughout a whole population or species.	Gene drives can be used to counter animal-borne diseases and can either arise naturally or be genetically engineered, e.g., using CRISPR (gene editing) technology.

**Table 3 biotech-10-00028-t003:** Codes and emerging themes from data.

SUPERORDINATE THEME	CODE AND SUBCODE
Attitudes towards the use of different genomic technologies	*Perception of the use of genomic technology*
	Genetic modification
	Gene drive
	Functional genomics
	Structural genomics
	Conservation genomics
	Proteomics
	Animal health and diseases
	Animal welfare
	*General concerns about the use of genomic technologies*
Prioritising the use of genomic technologies	*The relative importance of genomic technologies*
	Animal health
	Environmental sustainability
	Animal welfare
	Greater productivity
	Safer human food
	Efficient feed use
	Improved human wellbeing and health
	*Telos*
Ethical dilemmas from the use of genomic technologies	Animal welfare
	Animal health
	Free-range
	Concerns
	Religious concerns
	Naturalness
	*Telos*
Additional concerns	Climate change
	Organic vs. inorganic production
	Need for risk communication

## Data Availability

Data are available from the corresponding author on request.

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
