# Peer review of "Public Perceptions Regarding Genomic Technologies Applied to Breeding Farm Animals: A Qualitative Study"

_biotech, 2021, doi:10.3390/biotech10040028_

Round 1
Reviewer 1 Report
Dear Author, the paper is interesting, original and I have no particular remark to do. I noticed only some small misprint which needs to be corrected. Anyway, I am not qualified enough to judge about the English language and style
Line 317: be app0lied in the future: be applied
Line 431: …of the farm of animals: …of the farm animals
Line 522:… many participants perceived that that the application of some or all: … many participants perceived that the application of some or all
Line 527: … animal health and welfare, (see also [31–33]).: … animal health and welfare (see also [31–33]).
From line 549 to line 553: …The proponents….EU definition (36): this sentence is too long and not very clear and immediate to comprehend: please reformulate.
Line 558:..taken into account in discussed of the labelling…: ”in the discussion” or “when discussing the labelling”
Author Response
Reviewer 1.
Reviewer’s comments: Line 317: be app0lied in the future: be applied.
Authors’ response: Changed as requested.
Reviewer’s comments: Line 431: …of the farm of animals: …of the farm animals
Authors’ response: Changed as requested:
Reviewer’s comments: Line 522:… many participants perceived that that the application of some or all: … many participants perceived that the application of some or all
Authors’ response: Changed as requested:
Reviewer’s comments: From line 549 to line 553: …The proponents….EU definition (36): this sentence is too long and not very clear and immediate to comprehend: please reformulate.
Authors’ response: The text has been changed from line 579-582.
The proponents of modern gene editing techniques such as CRISPR-Cas9 argue that such techniques can be used to give additional or more complex types of genetic changes to those which would occur naturally. This should therefore be taken into account in legislative frameworks, for example in the EU definition
Reviewer’s comments: Line 558:..taken into account in discussed of the labelling…: ”in the discussion” or “when discussing the labelling”
Authors’ response: now removed as suggested by reviewer 3.

Reviewer 2 Report
Congratulations for conducting research in this area, which would be helpful to policy makers. It is important to learn about public opinion and make policies accordingly.
One of the major drawbacks I have noticed in this study is the results presented in this study are all opinion based. It is important to present results but scientific data analysis is very important to any research. The author mentioned in the materials and methods that a score was given by all the participants to all the genomic technologies but nothing was presented in the results. There could be several type of analysis can be performed using this data. I would strongly encourage to include that data in form of table or graphs and by gender, demographic area etc. These are ideas but authors should decided what they want to present and how they want their research study to be presented.
Some specific comments below-
Line 58: “.” Is not necessary before and.
Line 81: change “biotechnologies” to biotechnological methods
Line 134: is the protocol developed by researchers on this group or developed by some previous researcher. If it is from an previous study, please provide reference
LineL: 317: check spelling of applied
Reviewer 3 Report
Please see attached word document

Reviewer 4 Report
In this manuscript, the authors present their results of a focus-group study on the use of genomic technologies in farm animals. They found that participants were skeptical of invasive technologies motivated by financial interests. In contrast, tha application of genomic technologies for research purposes or to enhance animal welfare was seen positively.
The study addresses an interesting topic, and the manuscript provides a nuanced and interesting account of the results. However, the introduction to the topic and the connection to the general literature is rather rudimentary and should be complemented.
Specific comments
- Abstract, introduction: It sounds as if the study was only motivated by the aim to ensure consumer acceptance of animal-products. But as the authors mention at the end of the abstract, public priorities and attitudes should be considered (not only for financial reasons) in a democratic state. I think this point is actually more important and I would suggest mentioning it earlier to explain why we need such studies.
- L40 “At the same time, there is a body of evidence to suggest that public acceptance of the application of genomic technologies to animal production systems in nuanced by the type of genomic thechnology being applied….” You mention a body of evidence but then mention only one paper [9]. It would be important to give a more detailed account of what other authors found on the public opinion of genetic technology (even in plants) and ethics of genetic engineering in animals. In the discussion second you can then explain to what extent your data support, complement or disagree with the literature. I think in a qualitative study (which cannot and does not want to be representative) it is particularly important and interesting to discuss the literature in the field.
- L74ff: “While there is an extensive literature on public perceptions of, and attitudes towards genetic modification of plants, and to some extent to animals and micro-organisms, other areas of genomic science applied to agrifood production have not been so extensively researched» What do you mean by “other” here(not plants not animals and not microorganisms, what is left?)?
- P85: I like the separation into two aims (1. Exploring attitudes to different technologies 2. Discussing applications to animals) but I think against these background aims it is even more important to introduce the literature on attitudes towards genomic technologies in more detail at the beginning of the paper.
- I appreciate the detailed description of the study-methods, maybe it could even be shortened a bit (or put into a supplementary material section) in order to gain space for a more detailed introduction and discussion section.
- I think it is particularly interesting that you discuss not only engineering technologies but also “non-invasive” genomic technologies for research purposes. I think this might be a rather particular feature of this study (a comparison that other authors have not made) and could be highlighted more.
- Table 2: the example of the introduction of a “spinach gene” into pigs might have been slightly suggestive and trigger a “yuk-factor” response (thinking of green pigs…). It is not obvious, how a spinach gene in pigs can be useful, the gene for more and faster growing muscles in salmon might have been a more neutral example? But this is just a comment, I obviously do not suggest to remove this, as this is the information that your participants have received.
- P159: It could be helpful for some readers, if you could introduce the term “telos” for instance, with some reference to biocentric authors.
- P309: « Many participants expressed a preference for the application of genomic technologies to conserve animal species compared to other potential benefits that could be generated from the use of such technologies.» Could you explain in more detail, what kind of animal species would be conserved here? Were participants thinking of rare lifestock-species or did they not talk about the agricultural context?
- P10 and P11/12: I do not quite understand the difference between the “animal health and welfare” (p10) section and the “animal rights and welfare issues” (p11/12). I see that you discuss the first section as a benefit in the production system and the second as an ethical issue, but it seems to be the same point? Could you make the difference (if there is any) clearer or otherwise maybe combine the sections?
- 497: concerning Thompson: I actually think your data support Thompson’s point that people have moral intuitions against animal disenhancement (which is a very interesting observation), even though there are no strong philosophical arguments against it. Thompson is not saying animal disenhancement is ethically permitted or unproblematic, he speaks of a conundrum, because we do not have any strong rational arguments against animal disenhancement but nevertheless, most people have strong intuitions against it. There is a rich body of (ethics) literature on animal enhancement /disenhancement that you could discuss here.
- L503 «participants expressed the view that gene editing was to be preferred over and above genetic modification and the use of gene drives.» What exactly do you mean by “gene editing” how is it different from genetic modification? I assume you do not mean genome editing by CRISPR (as this could be more extensive than traditional gene technology and is the current technology used to introduce gene drives).
Round 2
Reviewer 2 Report
Thank you for addressing my comments.